# Effect of Photoperiod and Transfer Time on Atlantic Salmon Smolt Quality and Growth in Freshwater and Seawater Aquaculture Systems

Enrique Pino Martinez [1], Albert Kjartan Dagbjartarson Imsland [1,2,*], Anne-Camilla Diesen Hosfeld [3] and Sigurd Olav Handeland [1]

1   Department of Biological Sciences, University of Bergen, High Technology Centre, 5020 Bergen, Norway
2   Akvaplan-Niva, Iceland Office, Akralind 6, 201 Kópavogur, Iceland
3   Department of Environment and Aquaculture Technology, Faculty of Engineering, Bergen College University, 5020 Bergen, Norway
*   Correspondence: albert.imsland@uib.no or albert.imsland@akvaplan.niva.no

**Abstract:** Smoltification is a key process in Atlantic salmon aquaculture, given it prepares the fish for a successful transit from fresh to seawater. However, industry players have not yet reached a consensus on the best protocols to produce high-quality smolts. In this study, we assessed how the combination of two photoperiod regimes in freshwater (continuous light or LL, and natural photoperiod or LDN) and four transfer times to seawater (February, March, April, and May) affected smolt development and their subsequent growth in seawater until slaughter during commercial production. The results demonstrated that smoltification and growth in freshwater were only slightly modulated by the photoperiod treatment and were instead much more affected by the limiting effect of the low water temperature during that period. In seawater, the growth rate was the highest in the same groups, which had, however, experienced a delay in growth when in freshwater, and consequently, no differences in the final body weight between the eight treatments were found. Such compensatory growth in the sea was probably enhanced by the increasing smolt quality, which could allow for better performance in seawater. A significant link between the weight at slaughter and weight at transfer was observed only in the groups with a lower smolt quality (LL-Feb, LDN-Feb and LDN-Mar), which suggests that larger individuals could cope better with a saline environment. In contrast, smaller smolts probably suffered greater osmotic stress that hindered their performance at sea. Afterwards, as smolt quality increased in the subsequent transfer groups, the relevance of this size effect decreased. This means that the industry may benefit from transferring larger smolts to seawater, especially if these are suspected of having developed suboptimal seawater tolerance. Those individuals are likely to cope better with saline conditions than smaller smolts. Future research should focus on the possible long-term effects of freshwater-rearing regimes on smolt performance in the seawater phase.

**Keywords:** smoltification; salmon aquaculture; gill NKA activity; compensatory growth; specific growth rate

**Key Contribution:** The finding of the present study shows that smoltification and growth of Atlantic salmon in freshwater can be affected by the limiting effect of low water temperature. Smaller smolts will suffer greater osmotic stress that can hinder their performance at sea. Consequently, the salmon farming industry may benefit from transferring larger smolts to seawater.

## 1. Introduction

Smoltification is a crucial process in Atlantic salmon (*Salmo salar*) aquaculture, given that it prepares the fish to transit from the freshwater to the seawater phase [1–3]. During smoltification, salmon undergo a series of morphological, physiological and behavioral

changes that allow the fish to adapt to a marine environment. These include an increase in hypo-osmoregulatory abilities in organs, such as the gill, kidney, and gut, that result in high salinity tolerance, a reduction in the condition factor, a change in external coloration towards a silvery appearance, a darkening of the fin margins, and a switch from a territorial to schooling behavior among others [2–4]. In order to ensure optimal growth performance and fish welfare and to minimize mortality after a seawater transfer in aquaculture facilities, smolts must have developed optimal seawater tolerance [5]. Intensive farming of Atlantic salmon has become a cost-effective, season-independent industrialized animal production, with high throughput, reduced generation time, rapid development, high densities, and high-energy diets. Heated water and photoperiod control, with extensive use of constant light (LL), are used to accelerate the growth and development of season-independent smolt production and seawater transfer [5]. However, extended use of such extreme light regimes deprives the juvenile salmon of seasonal cues, critically interfering with the completion of parr–smolt transformation. It is thus crucial to identify the best rearing protocols that allow the production of such high-quality smolts and to develop tools to determine when the salmon are in optimal conditions to be transferred from fresh to seawater.

In nature, the smoltification process is synchronized by the naturally increasing day length in spring, with the water temperature acting as a rate-regulating factor [4,6–9]. In past years, before the introduction of environmental manipulation in aquaculture facilities, the release of farmed smolt was mainly limited to the period between April and June [10]. However, the extensive knowledge of smoltification gained over the years and the development of advanced technology solutions today allow the production of smolts all year round [5,11]. The most common method used to induce and accelerate the smoltification process comprises the use of photoperiod regimes that include a dark period or "winter signal" that is intended to mimic winter conditions, followed by a return to constant light in combination with an increased temperature, all during the freshwater phase [2,12,13]. However, empirical data from the industry shows that the quality of the smolts produced, measured in terms of survival and growth after transfer to the sea, is not optimal and varies between facilities (G. M. Knutsen, Bremnes Seashore AS, pers. comm.). Several producers claim that the reason for this variation is the difficulty in "timing" the optimal moment for seawater transfer in relation to the "smolt window", or the time period during which smolts have the best capacity to tolerate seawater [14,15]. This difficulty results from the faster smoltification rate and the higher average size that current smolts reach before they are transferred to seawater due to the mentioned intensification of rearing conditions.

Specific concerns on optimal transfer timing have been raised after observations taken at Sævareid Fiskeanlegg AS (T. Lohne, Sævareid Fiskeanlegg AS, pers. comm.), which have shown that the time the fish took to start eating was clearly dependent on the time of transfer to seawater. According to these data, the groups of fish that were released the earliest (March) took three months until 90% of the fish had food in their stomachs, while fish released a month later started eating after two months in the sea; this is at the same time as the first group. This resulted in a growth depression in the early group. The company then hypothesized that such lack of appetite was linked to the fact that the group was transferred too early, when fish had not yet completed smoltification, resulting in osmotic stress and underperformance.

Traditionally, one of the main markers of smoltification used in the aquaculture industry to determine the optimal time of transfer has been an increase in the gill Na+, K+, and ATPase activity (NKA), which allow salmon in seawater to excrete the excess sodium ions to maintain osmotic homeostasis [2,3,8,13]. This change in gill enzymatic activity is synchronized by the increasing day length that is typical of spring, indicating the development of hypo-osmoregulatory abilities that permit the adaptation to high salinity [3,7]. A different physiological marker of smoltification is the plasma concentration of monovalent ions, such as chloride and sodium, which tends to decrease during smoltification in freshwater up to the smolt window, followed by a transient peak after the transfer to seawater [16,17]. In regard to the morphological indicators of smoltification, a steady decrease in the condition factor

is among the most commonly used in aquaculture due to the simplicity of its measurement. This change in body shape results from the energetic demands of smolt development and from the change to a longer and leaner shape more suitable for life in the sea [2,3]. Finally, another remarkable external sign of smoltification is a change in coloration from greenish with dark marks typical of parr to a homogeneous silvery appearance that is more suited to pelagic life in a marine environment [3]. However, although these changes can be measured by staff in the rearing facilities, determining in absolute terms when they indicate the optimum time for seawater transfer remains challenging. Moreover, considering the current use of intensive light regimes that can result in the unsynchronized onset of smoltification among the farmed population [2], the determination of the best timing for smolt seawater transfer is key for the aquaculture sector.

Considering this, we designed the present study to investigate the effects of accelerating the smoltification process by using an advanced-phase light regime on smolt quality, development of seawater tolerance, and growth during the freshwater phase. In addition, this study examined the effects of transferring smolts to seawater before, during, and after the time of maximum seawater tolerance (the smolt window) on appetite and growth, both in the early post-smolt phase and until the fish were slaughtered at ~1.5 kg. With this, we aimed to increase the understanding of how the combination of both the transfer time (February, March, April, and May) and photoperiod treatment (constant light versus natural photoperiod) affect fish performance until slaughter. The generated knowledge could help identify which physiological and/or growth characteristics are best suited to predict the optimal smolt transfer time in the aquaculture industry.

## 2. Materials and Methods

### 2.1. Fish Stock and Early Rearing Conditions before the Experiment

This experiment was carried out at Sævareid Fiskeanlegg AS (FW facility) and a seawater facility at Høgsfjorden (Rogaland, Norway) in the period from 5 January 2016 to 15 February 2017. The first phase of the experiment, in freshwater until complete smoltification, was carried out at the freshwater facility and used 8000 farmed Atlantic salmon smolts (*Salmo salar* L.) from the Bolaks strain. The eggs hatched between 8 and 13 February 2015, and the first feeding started in early March, approximately 290 degree days (d °C) post-hatching at constant light and in heated water (14 °C). Thereafter, the fish were transferred to an indoor tank (2000 L) and reared at approximately 10–12 °C under a constant photoperiod from early March until 19 May 2015. Then, on 20 May, the fish were graded (average weight: 9.0 g) and transferred to two other tanks (with the same tank setup as above) and were provided with natural light (LDN) and ambient temperature conditions. During this initial period, the fish were fed a commercial standard dry diet according to the temperature and fish size (Skretting, Norway). In a growth measurement taken early in December 2015, the fish had an average size of 98.0 ± 11.9 g.

### 2.2. Experimental Design

In early January 2016, the fish were distributed into 16 outdoor rearing tanks (2000 L) containing 500 fish each. A representative subpopulation of these pre-smolts was individually marked with PIT tags for further studies of growth in the sea. Tagged fish were randomly distributed among the 16 experimental tanks (*n* = 50 per tank). One week after the fish distribution, a 2 × 4 factorial design was established (Figure 1), including two photoperiods (a constant photoperiod or LL, and a natural photoperiod or LDN, 60°25′N) and four different times for sea transfer (February, March, April, and May). The eight resulting groups (LDN-Feb, LDN-March, LDN-Apr, LDN-May, LL-Feb, LL-March, LL-Apr, and LL-May) were run in duplicates. All tanks were covered with a tarpaulin to prevent light penetration between the groups. The light was provided using fluorescent tubes mounted under the tank cover, and light regimes were maintained until the fish were transferred to seawater. In freshwater, the fish were fed according to the Skretting feeding tables, from 07:00 to 15:00, using the suspended feed Nutra smolt 3 mm, with daily adjustments in

relation to appetite and temperature. The temperature in the water was recorded daily during the entire freshwater phase and in seawater until November 2016 (Figure 2). The oxygen saturation in the outlet water remained over 80% during the whole experimental period. When the freshwater phase ended in the respective groups after week 8 (Feb), week 12 (March), week 17 (April), and week 20 (May), the fish from both photoperiod groups (LL and LDN) were transported from the freshwater facility to a standard 5 × 5 m open pen in seawater in a common garden setup. In the seawater, all groups were fed in excess with Nutra smolt 3 mm containing X-ray balls (see the description below). All groups were reared according to standard routines and were on ordinary feed to a size of approximately 1.5 kg. The experiment was terminated on 15 February 2017.

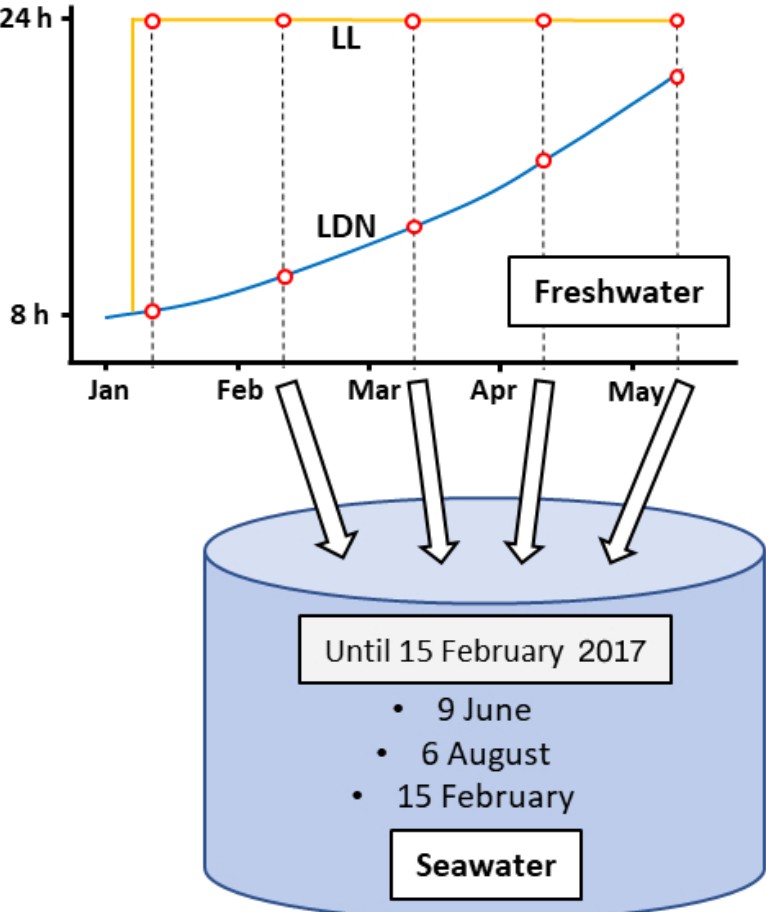

**Figure 1.** Diagram of the experimental setup. During the period in freshwater, the fish were reared under either continuous light (LL) or natural photoperiod (LDN). Both groups were sampled once a month (represented by red circles) and challenged with a 24-h full seawater challenge test. After this, fish from both photoperiods were transferred to seawater in February, March, April, and May. Samplings in seawater took place on 9 June and 6 August (2016) and 15 February 2017, when the experiment was terminated.

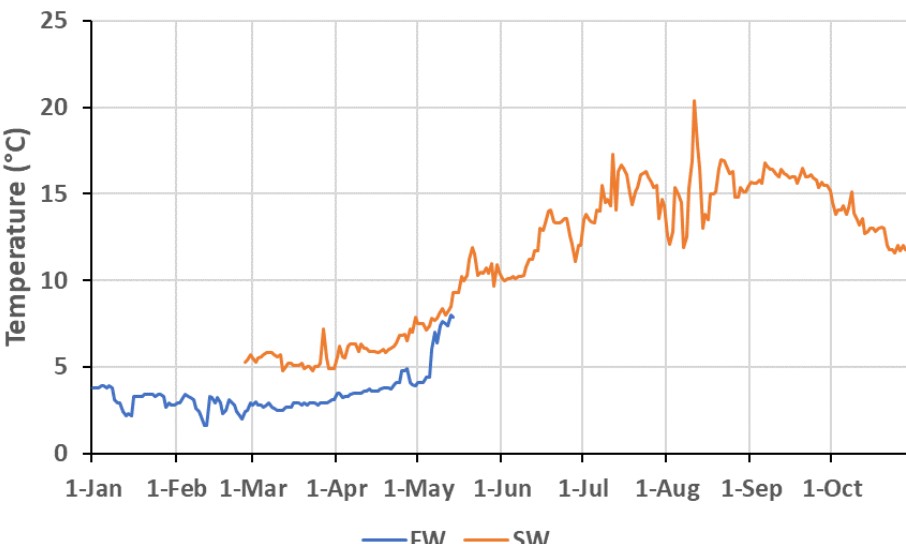

**Figure 2.** Temperature (°C) in freshwater (in blue) from 1 January to 15 May 2016 and in seawater (orange) from 1 March until the end of October 2016.

*2.3. Sampling Procedures and Analyses*

Throughout the freshwater period, physiological samples were collected from the unmarked part of the population. The purpose of these samplings was to monitor the smolt quality during the parr–smolt transformation in order to relate it to the time of transfer to the sea and further growth in seawater. The first sampling was performed in January when the fish were divided into different photoperiods (LL and LDN). Afterwards, the samples were collected monthly until the individual groups were transferred to the seawater (February, March, April, and May). All groups were starved 24 h prior to sampling, and each time 20 fish were removed from each tank and killed by a blow to the head. Blood was collected with heparinized syringes from the caudal peduncle, and plasma was obtained by centrifugation at 4000 rpm and at 4 °C and analyzed for chloride levels (mM) in duplicate 20 μL samples using a Radiometer CMT 10 titrator. Gill filaments were dissected, frozen in SEI buffer at −80 °C and subsequently analyzed for NKA activity using the method of [18]. In addition, the hypo-osmoregulatory ability was assessed for every transfer time by measuring the plasma chloride levels after a 24-h seawater challenge test (34.5%). The weight (to the nearest 0.1 g) and length (to the nearest 0.1 cm) of the smolts were determined to study the individual growth. The individually marked part of the population was followed up in the sea with the registration of their length and weight. The specific growth rate (% per day) was calculated as SGR = (ln(W2) − ln (W1)) × 100/(T2 − T1), where W2 and W1 are the weights in consecutive times (g), and T2-T1 is the number of days between the two measurements. The condition factor was calculated as K = weight × 100/(length)$^3$. Food in the stomach was assessed 2, 4, and 6 weeks after the transfer to seawater (February, March, April, and May). This was performed by replacing the ordinary diet with an identical diet containing X-ray-dense Ballotini glass beads prepared by Skretting (Stavanger, Norway). The new feed had an identical nutrient composition, color, particle size, and texture as the normal commercial diet, but it allowed the measurement of the individual food intake rates and stomach evacuation by X-radiography (GeR XT-100 X-ray machine, AGFA Structurix DX7, [19]). Following the development of the X-ray plates, the number of marker particles present in the stomach was counted, and the amount of feed consumed was calculated. This calculation was performed with a standard curve prepared by X-raying the known weights of the marked food and assessing the number of Ballotini glass beads. With this information, a linear regression between the weight of the marked feed and the number of Ballotini particles was carried out and used as a standard curve as follows:

$$\text{Food intake (mg dry feed/g wet fish)} = 0.05 \times \text{(Ballotini glass beads)} + 0.05, (N = 12, r^2 = 0.97) \tag{1}$$

Following the transfer to seawater, the fork length (to the nearest 0.1 cm) and weight (to the nearest 0.1 g) of the individually tagged salmon post-smolts were measured on 9 June, 6 August, and 15 February.

### 2.4. Statistical Analyses

All statistical analyses were performed in R and Rstudio, using the packages "car" [20], "ggplot2" [21], "ggpubr" [22], "Rmisc" [23], "emmeans" [24], and "nlme" [25]. For the freshwater stage, possible differences between the experimental groups were investigated using a two-way nested ANOVA with the predictors "Photoperiod" and "Date" and their two-way interaction as fixed effects, and "Tank" as a random effect. For the seawater stage, possible differences between the experimental groups were investigated using a three-way ANOVA with the predictors "Photoperiod", "Transfer Time", and "Date" and their two-way interactions as fixed effects. Possible differences in the SGR in individually tagged fish after their transfer to SW were investigated with a two-way nested ANOVA with "Photoperiod" and "Transfer time" with their two-way interactions as fixed effects and "Tank" as the random effects. Prior to fitting all statistical models, the distribution and existence of outliers in the response were checked with the Shapiro–Wilks test and boxplots, respectively. The homogeneity of variance was checked with Levene's test. In cases of significant ANOVAs, Tukey HSD post hoc tests were run to find significant differences in the response variable between pairwise groups at each sampling and within the experimental groups over time. Multiple regression was performed between the weight at transfer (g) and weight at slaughter (g) for each transfer group and photoperiod in order to assess if the weight of the salmon at slaughter was linked to their size at the time of their seawater transfer under the different experimental conditions. Plots of all variables display the mean ± standard error of the mean (SEM). A significance level of $\alpha = 0.05$ was always used.

### 2.5. Ethic Statement

The ethical policies of the journal were followed. The study was approved by the local representative of Animal Welfare at the Department of Biological Sciences, University of Bergen, Norway (FOTS application ID8017), and the samplings were performed as established by the Norwegian Animal Research Authority.

## 3. Results

### 3.1. During Smoltification in FW

#### 3.1.1. Body Weight and CF

The body weight (Figure 3A) was only significantly dependent upon the time ($p < 0.001$). Over time, a significant increase in weight was only present in LL from January to April ($p < 0.01$) but not in the LDN. No significant differences in weight occurred between LL and the LDN for any sampling.

The condition factor (Figure 3B) was also only dependent on time ($p < 0.001$). Over time, this variable decreased in parallel in both photoperiod groups from January to April ($p < 0.01$). No significant differences occurred between LL and the LDN at any sampling.

#### 3.1.2. Gill NKA Activity and Plasma Chloride in FW and in 24-h SW Challenge

The gill NKA activity (Figure 4A) was significantly dependent only on time ($p < 0.001$). Significant increases in this enzyme activity occurred in both photoperiod groups in parallel, from January to February (both $p < 0.001$) and from March to April (both $p < 0.001$). NKA continued to increase in the LDN group until May ($p < 0.001$) but not in LL. No differences in the gill NKA were observed between LL and the LDN at any time.

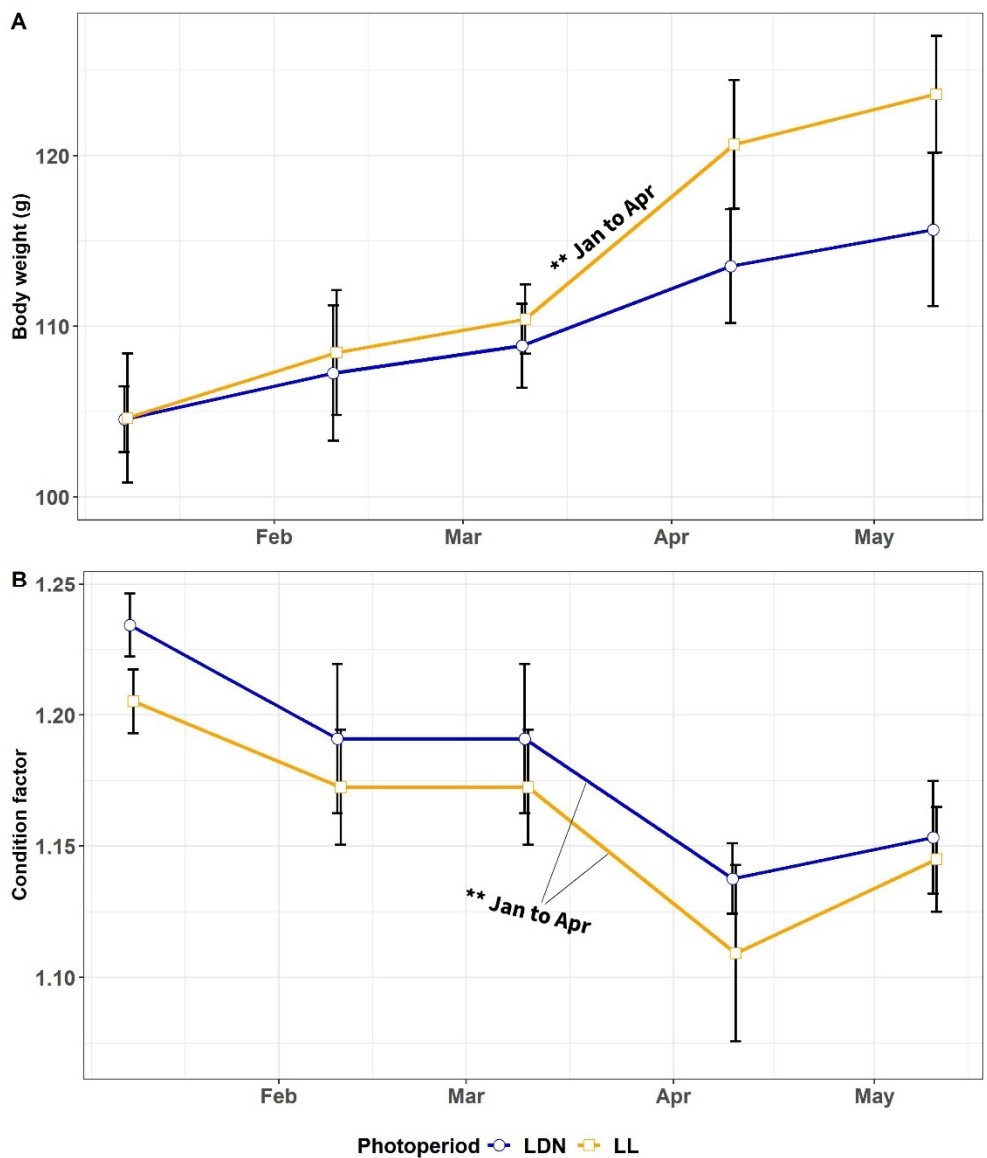

**Figure 3.** Body weight in g (**A**) and condition factor (**B**) over time in both photoperiod groups during smoltification in FW. Asterisks represent significant differences in both variables between samplings, and ** indicates that *p* < 0.01 (two-way nested ANOVA). Vertical lines indicate SEM.

The plasma chloride concentration (Figure 4B) was dependent on the time and water quality (both *p* < 0.001) but not on the photoperiod. In freshwater, parallel decreases in chloride were observed in both photoperiod groups from January to April (both *p* < 0.001), but this decrease continued only in LL until May (*p* < 0.001). As a result, the chloride levels were higher in the LDN than in LL in May (*p* < 0.05). The plasma chloride levels were significantly higher after all 24-h SW challenge tests in all months and photoperiods (all *p* < 0.001). The highest plasma chloride levels of any seawater challenge test occurred in February in both photoperiod groups, decreased until March (both *p* < 0.001), and became stable afterwards. The only difference between the photoperiod groups in a chloride concentration during the seawater challenges occurred in February, which was when the plasma chloride was higher in the LDN than LL (*p* < 0.05).

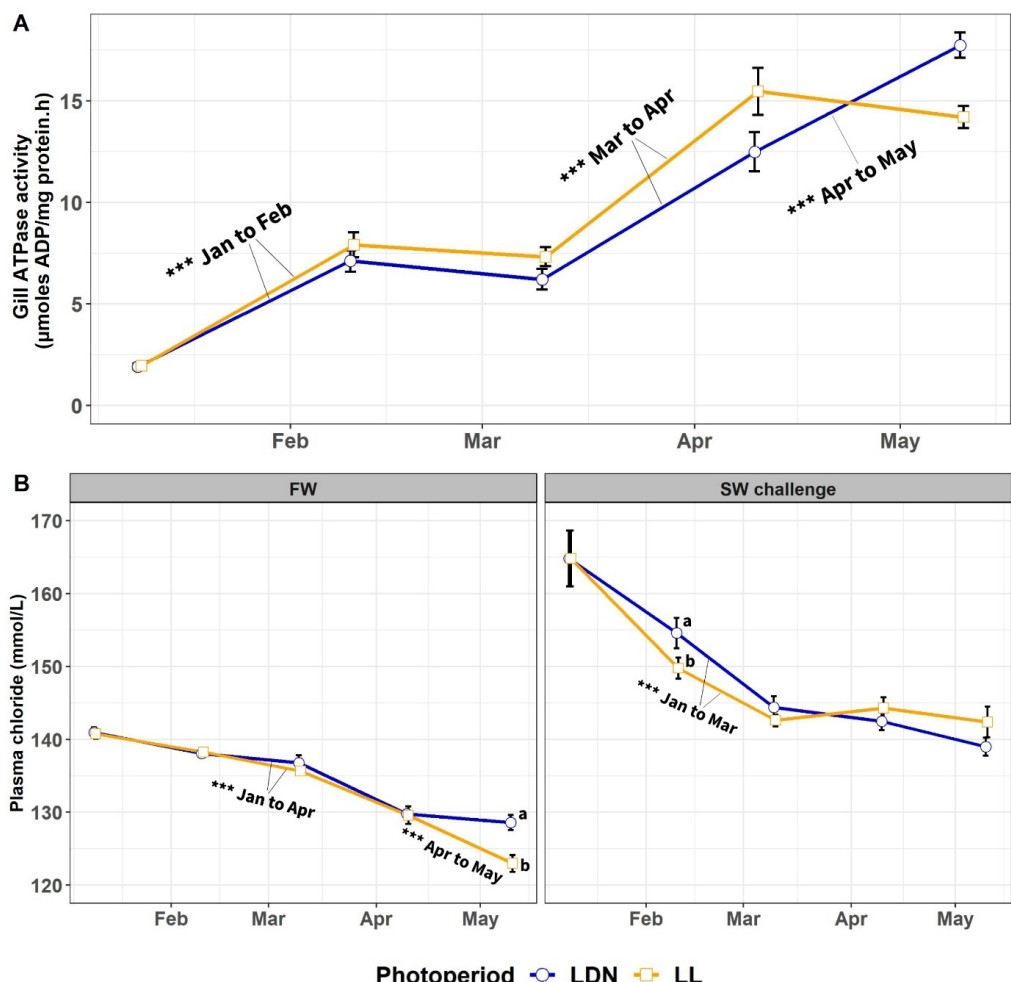

**Figure 4.** Gill NKA activity (**A**) and plasma chloride concentration in freshwater and after a 24-h sea water challenge (**B**) in both photoperiod groups during smoltification. Asterisks represent significant differences over time, indicated by *** ($p < 0.001$, two-way nested ANOVA). Letters "a" and "b" represent significant differences ($p < 0.05$) between photoperiod groups at a given time (Tukey post hoc tests). Vertical lines indicate SEM.

*3.2. After Transfer to SW*

3.2.1. Body Weight and Mortality

The body weight after the transfer to seawater (Figure 5) was significantly dependent upon the photoperiod, transfer time, date (all $p < 0.001$), and interactions photoperiod × date ($p < 0.001$) and transfer time × date ($p < 0.01$). Significant differences in weight between the transfer groups occurred in fish from both photoperiods but were more pronounced in the LDN groups. Thus, the fish transferred in February were generally significantly heavier than the other transfer groups in June and August. All transfer groups from LL (except those transferred in February) were heavier than their LDN counterparts in June and August (all $p < 0.05$). However, despite these early differences, no differences in body weight occurred between transfer groups or photoperiods at the time of slaughter in February.

Mortality only occurred in seawater in the groups transferred in February, but with a clear difference depending on the photoperiod (a 10% mortality in the LDN compared to 1% in LL).

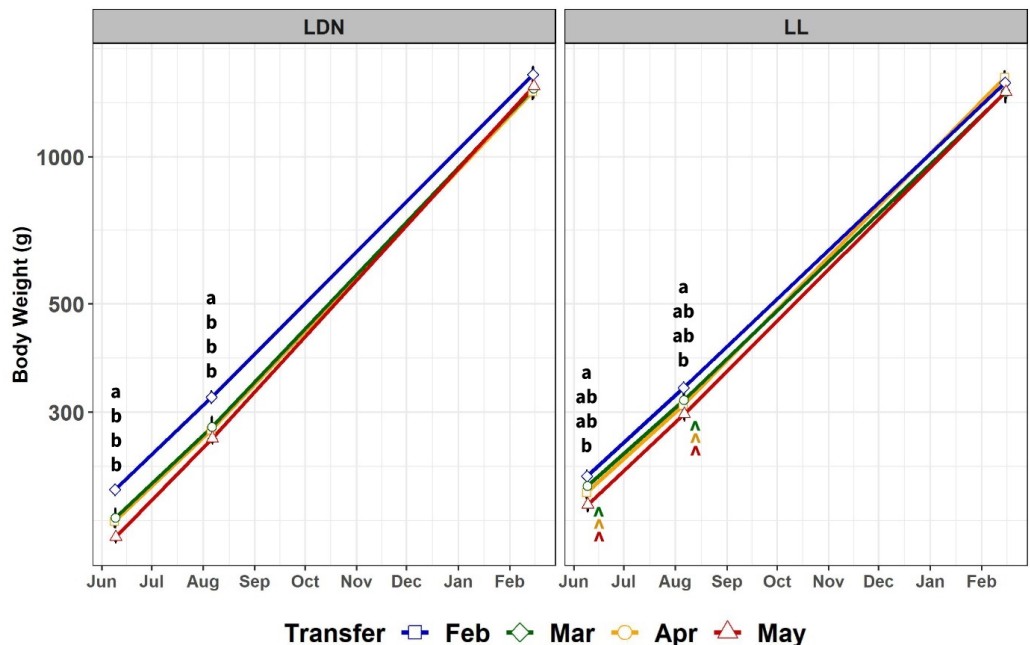

**Figure 5.** Body weight in the different transfer and photoperiod groups during the period they were all in seawater. The "Y" axis is displayed on a logarithmic scale for better visualization. Photoperiod regimes are displayed separately, with LDN groups on the left and LL groups on the right. Letters "a" and "b" indicate significant differences ($p < 0.05$, Tukey post hoc test) at a given sampling between groups transferred to seawater at different months within each photoperiod regime. Colored signs "^" indicate significant differences ($p < 0.05$, three-way nested ANOVA) between groups transferred at the same time but that experienced different photoperiods. These signs are located next to the largest of the pair and have the color of the transfer group. Vertical lines indicate SEM.

3.2.2. SGR of Individually Tagged Fish after all Periods in SW

The SGR (% body weight/day) during the period in SW (Figure 6) was significantly dependent on the photoperiod ($p < 0.001$) and transfer time ($p < 0.01$). The SGR was generally greater in the LDN than in LL, especially in March ($p < 0.05$) and May ($p < 0.01$). Within the LDN groups, the SGR was larger in the fish transferred in May than in those transferred in February ($p < 0.01$). No differences in the SGR occurred within the LL groups transferred at different times.

3.2.3. Food in Stomach after 2, 4 and 6 Weeks in SW

Food in the stomach was dependent on the photoperiod, transfer time, date (all $p < 0.001$), and interactions photoperiod × transfer time ($p < 0.001$) and transfer time × date ($p < 0.01$). The amount of food in the stomach (Figure 7) was larger in the groups transferred to SW in April and May than in the fish transferred in February and March in all sampling times under both photoperiods. However, in some cases, groups transferred to SW in the same month contained more food in the stomach if they had been reared under the LL photoperiod (the May group after 2 and 4 weeks in SW, both with $p < 0.05$, or the March group after 4 weeks in SW, $p < 0.05$). Over time, all eight experimental groups showed a significant increase in the amount of food in the stomach (all $p < 0.001$).

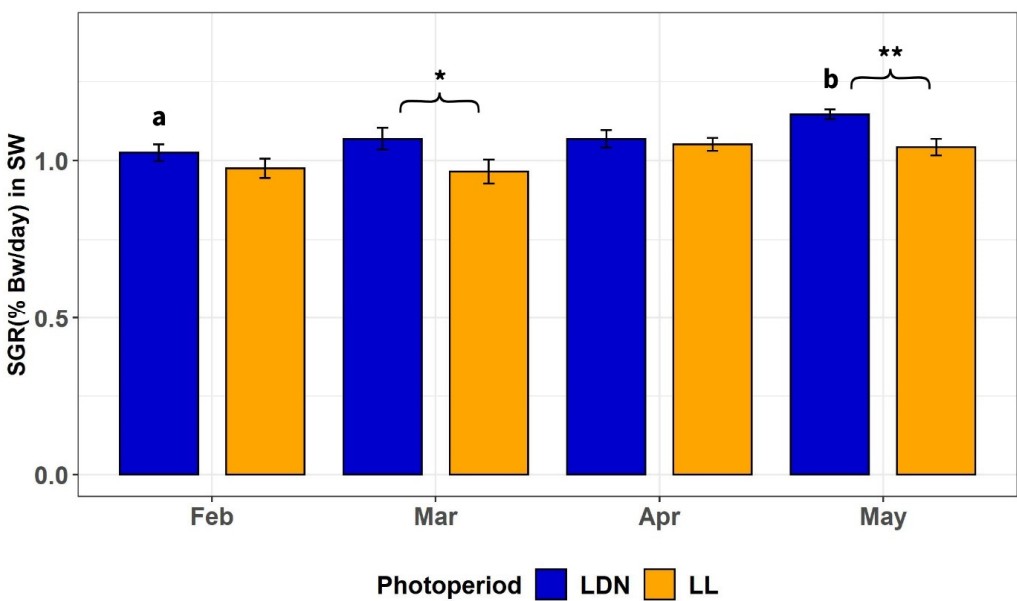

**Figure 6.** The specific growth rate in seawater (SGR, % body weight gain per day) of individually tagged fish that had been reared under LDN or LL photoperiods and transferred to seawater at different months. Asterisks represent significant differences (two-way nested ANOVA) between photoperiod groups transferred in the same month, as follows: (*) $p < 0.05$; (**) $p < 0.01$. Letters "a" and "b" indicate significant differences (Tukey post hoc test, $p < 0.05$) between groups reared under the same photoperiod but transferred to seawater at different times. Vertical lines indicate SEM.

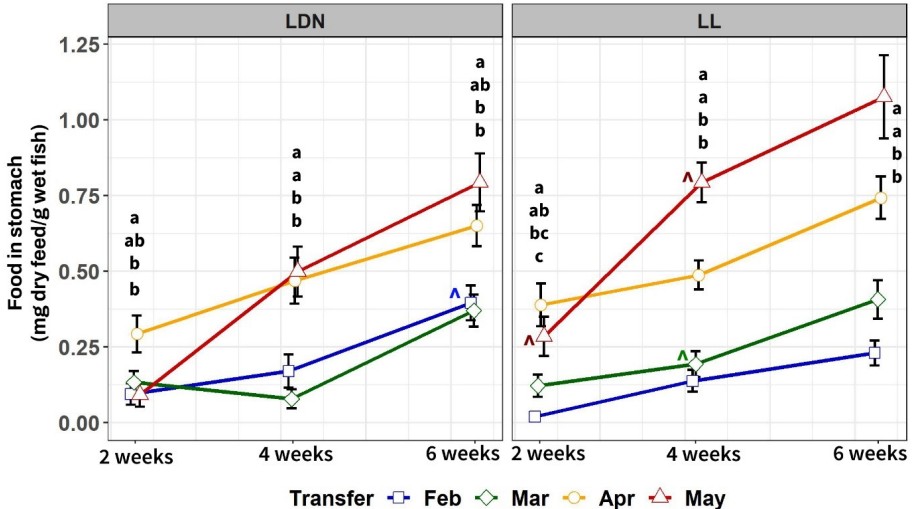

**Figure 7.** Food in stomach (mg of dry feed per gram of wet fish) after 2, 4, and 6 weeks in seawater, in groups of fish transferred to seawater in different months and reared under LDN or LL photoperiods. Letters "a", "b", and "c" indicate significant differences (Tukey post hoc test, $p < 0.05$) at a given sampling between groups transferred to seawater at different months within a photoperiod regime. Colored signs "^" indicate significant differences (three-way nested ANOVA, $p < 0.05$) between groups transferred at the same time but that experienced different photoperiods and are located next to the largest of the pair in the color of the transfer group.

### 3.2.4. Size at Transfer to SW vs. Final Size at Slaughter

Since the statistical model in 2.1 had revealed significant effects on the body weight of transfer time, date, and of the interactions photoperiod × date and transfer time × date, we investigated a possible link between the weight at transfer and weight at slaughter

by performing linear regressions between these two variables for each transfer group separated by a photoperiod. The results showed that a larger weight at transfer was linked to a larger size at slaughter in only the fish transferred in February reared in LL ($p < 0.01$, $R^2 = 0.094$) and the LDN ($p < 0.01$, $R^2 = 0.162$) and in fish transferred in March reared under the LDN ($p < 0.01$, $R^2 = 0.172$), but not in the rest of the groups (Figure 8, see the other regression parameters in the figure caption).

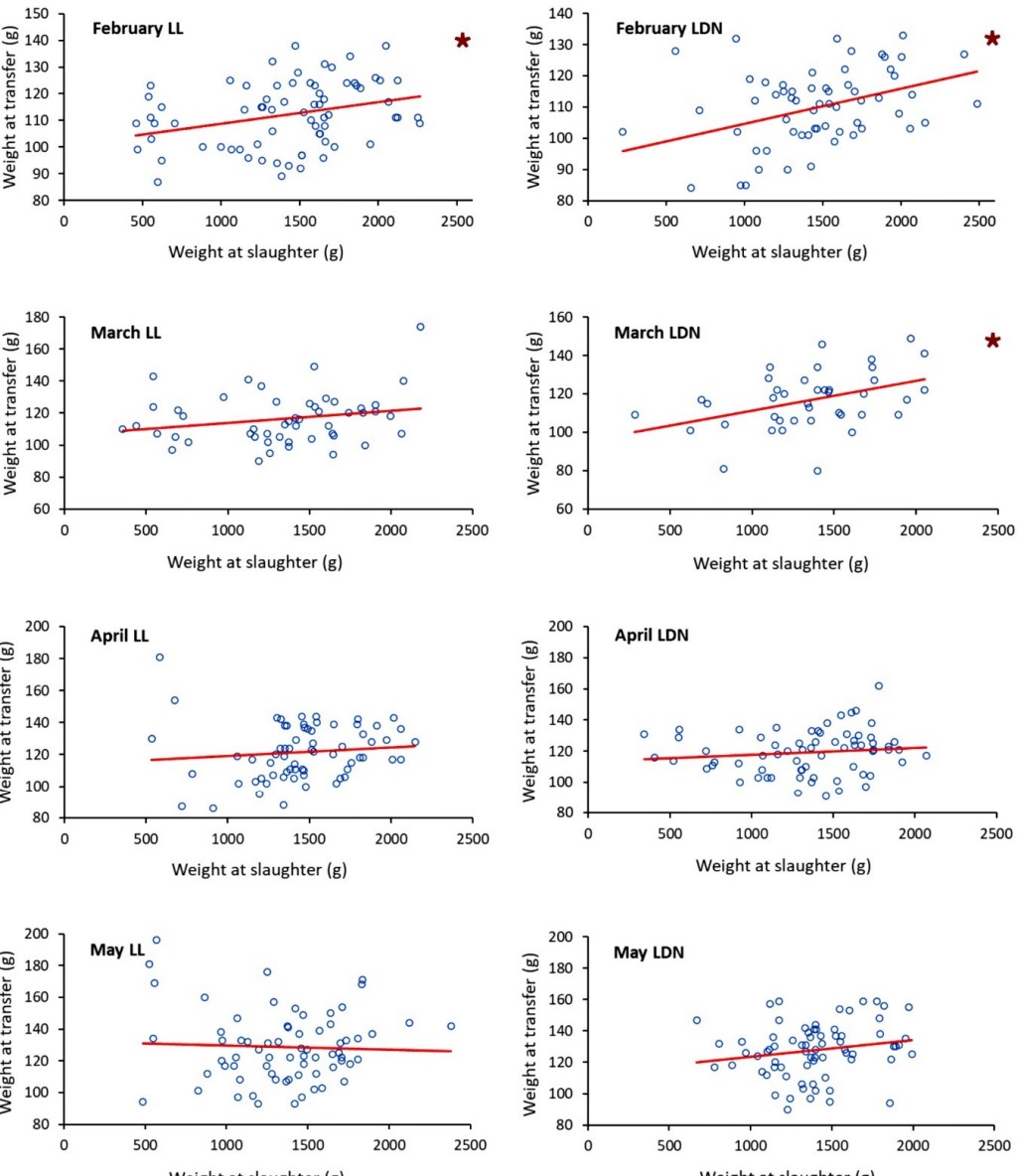

**Figure 8.** Linear regressions between weight at transfer and weight at slaughter (both in g) for groups of fish transferred to seawater in February, March, April, and May and reared in freshwater either under constant (LL) or natural (LDN) photoperiods. The regression parameters for each case are as follows: February LL ($p < 0.01$, $R^2 = 0.094$), February LDN ($p < 0.01$, $R^2 = 0.162$), March LL ($p = 0.120$, $R^2 = 0.049$), March LDN ($p < 0.01$, $R^2 = 0.172$), April LL ($p = 0.384$, $R^2 = 0.012$), April LDN ($p = 0.301$, $R^2 = 0.016$), May LL ($p = 0.735$, $R^2 = 0.002$), and May LDN ($p = 0.117$, $R^2 = 0.035$). Asterisks indicate regressions with *p*-value $< 0.05$.

## 4. Discussion

### 4.1. Growth and Smoltification in Freshwater

The photoperiod regime did not highly influence growth during the freshwater stage. This is evidenced by the lack of significant differences in body weight between the photoperiod groups at any sampling. However, the significant increase observed in the body weight over time in LL but not in the LDN indicates a higher growth rate in LL in response to early exposure to continuous light. The enhancing effects of additional light on salmon growth have been largely documented in the past [26–29] and linked to growth hormone production increases in response to constant light [8,30]. Furthermore, it has been estimated that one month's difference in exposure to constant light can improve growth between 13 to 20% [31]. In our study, as mentioned, this effect was weak, with both photoperiod groups displaying low growth and only differing in a more pronounced increase over time in the body weight in LL than in the LDN. This is most likely explained by the very low water temperature experienced by both photoperiod groups during the freshwater phase (see Figure 2), with a mean temperature of 3.2 ± 0.6 °C between 1 Jan and 5 May. Water temperature acts as a rate-controlling factor for life history traits, such as growth [28] and smoltification [32] in Atlantic salmon, modulating the effects of the photoperiod. The water temperature, to optimize growth and FCR in salmon, has been estimated at approximately 12–14 °C for fish of 70–300 g [33], which is, by far, higher than the temperature experienced by our groups. As such, the positive effect of constant light on salmon growth is likely to be restrained at a temperature as low as 3 to 4 °C.

Similarly, smoltification was not largely affected by the photoperiod treatment. Sometimes, continuous light has been found to negatively affect the development of hypo-osmoregulatory abilities compared to using a reduced photoperiod regime or winter signal [7,11]. However, typical signs of smoltification, such as an increase in gill NKA activity, a reduction in the condition factor, and a decrease in plasma chloride levels [2,11], occurred in parallel in both photoperiod treatments, with hardly any significant differences between the groups in any samplings. Furthermore, the plasma chloride levels after the 24-h seawater challenge tests also displayed similar decreases over time under both light regimes. Together, these results indicate that smoltification progressed similarly in both groups, irrespective of the photoperiod experienced. This is in line with previous research, concluding that smoltification is likely to commence in salmon simply by size, as part of the fish's continuous development, independently of the photoperiod [11,28,34,35]. According to [28], smolt characteristics, such as elevated gill NKA activity, are observed in salmon as they reach a certain size range (113–162 g), regardless of the growing conditions. Similarly, [34] concluded that a photoperiod cue is unnecessary for aquaculture strains of Atlantic salmon to initiate smoltification. The occurrence of size-induced smoltification has implications for modern aquaculture. Hypothetically, salmon of 150–200 g, which are kept in freshwater, may have previously acquired smolt characteristics, but if not exposed to seawater, they will de-smoltify, partially losing the hypo-osmoregulatory abilities developed [1,2]. Although de-smolted salmon have shown a rapid capacity to adapt to abrupt changes in salinity [34], the risk of poor performance and higher mortality of such large de-smolted post-smolts in seawater could be higher.

As occurred with growth, the very low water temperature during the freshwater phase could well explain the absence of clear photoperiod effects on smoltification. Water temperature does not act as a zeitgeber or environmental cue to synchronize the onset of smoltification, as increasing day length does [9]. However, water temperature determines the developmental potential of the fish and the rate of response to photoperiod cues, consequently affecting the rapidity and intensity of initiation and development of the smoltification process [9,30]. Indeed, a high water temperature has been found to advance the parr–smolt transformation and the rate of smolt development until reaching the "smolt window" or an optimum stage for seawater readiness [9,14,28]. Based on this, if smoltification occurs under very low water temperatures, as in our study, the rate at which the process progresses must be necessarily reduced, and thus, the role of the photoperiod

as zeitgeber could be masked. This is aligned with the results from [32], who reported that salmon reared at very low water temperatures (similar to the one in our study) displayed a delayed and less intense response to increased day length than those reared at 10 °C, which was observed in several endocrine and physiological markers of smoltification. The authors thus concluded that a low water temperature limits the physiological response to an increased day length associated with smoltification, such as gill NKA activity, plasma growth hormone, or plasma insulin-like growth factor I, among others. In the context of our study, the low temperature may have limited the influence of the photoperiod on smoltification to an extent sufficient to impair significant differences between the photoperiod treatments in the indicators of smoltification analyzed. Then, the temperature increase occurring from mid-March, and especially in May (2.5–8 °C), probably contributed to accelerating smoltification in both photoperiod groups since increasing temperatures have been found to contribute to the development of seawater tolerance in salmon [36].

As a result, signs of smoltification were obvious in both photoperiod groups in freshwater in April and May, when salmon seemed to reach their optimum stage for seawater transfer (smolt window). This signals that, regardless of the photoperiod, the smolt quality was higher in April and May than in February and March. Consequently, it may be expected that salmon transferred in April–May would have better performance in terms of growth and mortality during the seawater phase and until slaughter than those transferred in February and March.

### 4.2. Body Weight, Growth Rate, Mortality, and Feed Intake after Transfer to Seawater

After seawater transfer, the body weight of the groups moved in February initially increased faster than the rest, but as time progressed, the groups transferred in May, which was initially the most delayed, compensated for this effect. Regarding the photoperiod groups, all treatments that had experienced LL were initially larger in seawater, but later, those in the LDN compensated for this. The final result was that, at slaughter, all eight treatments displayed no differences in body weight, as the compensating effect in the transfer and photoperiod groups had eliminated the differences in body weight found earlier in the trial period. Accordingly, the SGR of the individually tagged fish in seawater generally showed an increasing trend in the subsequently transferred groups and tended to be higher in the LDN than in the LL groups. Two main aspects combined can explain these observations: a limitation on growth imposed by the colder freshwater temperature in the groups transferred to seawater later and the compensatory growth after being adapted to seawater.

First, the groups that were transferred to seawater earlier experienced a remarkably higher water temperature than those that remained in freshwater for a longer time, as groups were sequentially moved from fresh to seawater. Recalling the temperature effect on the growth widely discussed in the previous section, the longer exposure to a low temperature of the groups transferred in May must have restrained their capacity for growth, in contrast with the groups transferred in February or March. The photoperiod regime experienced during the freshwater phase also influenced the body weight in seawater, but only during the first stage, after which these differences were diluted, resulting in all groups being similar at slaughter. This is probably the result of the trend observed in freshwater, where the LL groups tended to be larger each month than the LDN groups. It is well known that exposure to continuous light can increase growth in salmon [28], but in our case, probably, the overall effect of the low temperature reduced the potential differences caused by different photoperiods [32]. However, we noticed a relevant difference between the photoperiod groups in the mortality of smolts transferred in February, with a 10% mortality in the LDN group compared to only 1% in the LL group. The presence of mortality in the groups transferred in February and not in the rest may suggest an overall lower seawater readiness of all groups in February, which is consistent with the results on the smoltification indicators previously discussed. However, the clear difference in mortality between the LL and LDN treatments in February suggests that the

phase-advanced light regime probably induced a larger capacity to tolerate high salinity conditions in LL vs. LDN already in February, despite none of the photoperiod groups being yet in optimal conditions. This is consistent with the previous research reporting the benefits of experiencing a phase-advanced photoperiod regime vs. a natural photoperiod for the earlier development of seawater tolerance [2,8,12,30,37].

Second, subsequently transferred treatments experienced compensatory growth in seawater that was most obvious in the groups transferred in May. These groups had displayed a delay in growth in freshwater due to the low mean water temperature but compensated for such a delay by growing at the fastest rate in seawater, resulting in no differences in body weight at slaughter on 15 February 2017. Compensatory growth can occur in salmon after periods of poor conditions for growth, such as low temperature, reduced light, or low availability of food [38–40]. This mechanism allows fish to make efficient use of periods with good opportunities for growth to restore their energy reserves that had been depleted during periods with more challenging environmental conditions [38]. Compensatory growth can be more pronounced after a change to higher water temperatures [39], such as, for example, the one experienced in seawater by the groups transferred in May. Another important factor that most likely contributed to the largest compensatory growth in the May groups is the general improvement in smolt quality that we observed from February to May. All condition factors, seawater tests, behavior, and gill NKA activity data showed a clear tendency towards improved smolt quality in such a period, and this is positively correlated to the observed long-term growth rates in seawater. In addition, it has been shown that smolts can often experience a temporary increase in their growth rate in the first 4–6 weeks in seawater [41], which would help to explain the effect in all transfer groups. Compensatory growth in the sea was also modulated by the photoperiod regime experienced in freshwater, being less pronounced in the LL than in the LDN groups. This suggests that the slight delay in growth caused by the LDN in freshwater by the reduced exposure to light [28] was later compensated in seawater to a higher extent in those groups. Similar findings were reported by [11], who linked better seawater growth to having experienced a natural photoperiod and better smolt status, which might also be the case in our study.

The pattern observed in the feed intake after 2, 4, and 6 weeks in seawater is clearly linked to the different sizes and temperatures at which each group was transferred. The salmon transferred in May had larger sizes and were moved to a higher temperature than the groups transferred in February, March, and April. Appetite and feed intake are known to increase with water temperature and body weight [28,33], and as such, it could be expected that the groups transferred in April and May contained more food in their stomach after 2, 4, and 6 weeks in seawater than those groups transferred in February and March.

### 4.3. Relationship between Weight at Transfer and Weight at Slaughter

We found that the body weight at slaughter was significantly and positively correlated with the body weight at transfer only in fish transferred in February (both LL and LDN groups) and March (LDN group) but not in the rest. This can be linked to the increasing smolt quality observed in the subsequent transfer times but also suggests that smolt size matters for their performance in seawater in cases where smolts have not yet reached their optimum peak smolt window. Thus, all individuals in February and the LDN group in March had probably developed the least seawater tolerance among all groups, and as a result, their performance in the sea benefited the most from having reached a larger body size. It is known that body size is a primary determinant for the parr–smolt transformation and increased salinity tolerance in Atlantic salmon [1,2], and individuals with larger sizes show better osmoregulatory performance in seawater challenge tests than smaller individuals [6]. At a larger size, salmon display a more favorable relationship between body area and volume and thus, maintaining osmotic homeostasis in a saline environment becomes less challenging [1]. In contrast, the smallest individuals will experience greater

osmotic stress in seawater and, thus, are more likely to show poorer performance during the first time in the sea compared to large fish. Considering this, it is likely that, in our study, the individuals transferred to the sea earlier (February) suffered greater osmotic stress due to their smaller size; thus, larger individuals within the group benefited from a better capacity to cope with higher salinity, thus performing better and reaching larger sizes at slaughter. However, the relevance of this effect will most likely decrease as the smolt quality increases (the groups transferred in April and May); thus, in these groups, the size of the smolts at transfer will not be as relevant for their performance at sea and, consequently, will not be linked to the size they reach at the time of slaughter. An implication for the producers is that, in case of uncertainties related to the smolt quality before transferring a commercial batch to seawater, the producer may benefit from doing so with a larger fish size, maximizing the capacity to cope with the osmotic challenges in seawater.

## 5. Conclusions

Smoltification and growth in freshwater were only slightly modulated by the photoperiod treatment and, instead, much more determined by the limiting effect of the low water temperature during that period. As a result, smoltification progressed similarly over time under both light regimes, a process probably enhanced by the relatively large size attained by the fish already in January and by an increase in temperature from mid-March. As a result, the smolt quality was the highest at the end of the freshwater phase in April and May, largely irrespective of the photoperiod. In seawater, the growth rate was higher in the groups transferred later, which had experienced a delay in growth in freshwater. This compensatory growth, observed in the sea, resulted from such limited growth in freshwater and the increasing smolt quality in subsequently transferred groups, which must have permitted better performance in the seawater. The link between the weight at slaughter and weight at transfer in the groups transferred in February (LL and LDN) and March (LDN) suggests that smaller smolts suffered greater osmotic stress, thus showing poorer performance at sea, while larger smolts benefited from their larger size to cope better with a saline environment. However, the importance of this size effect decreased as the smolt quality increased in subsequent transfer groups.

**Author Contributions:** S.O.H. and A.K.D.I. established the project, gathered the funding, and designed the experiment. A.-C.D.H. and S.O.H. carried out the samplings. E.P.M. performed the NKA analyses, and A.-C.D.H. analyzed the plasma chloride. E.P.M. and S.O.H. carried out data analysis. E.P.M. drafted and wrote the manuscript. A.-C.D.H., S.O.H. and A.K.D.I. provided editorial assistance and helped write the document. All authors have read and agreed to the published version of the manuscript.

**Funding:** This study was fully financed by Sævareid Fiskeanlegg AS (5645, Hordaland, Norway).

**Institutional Review Board Statement:** The present field trials were approved by the local responsible laboratory animal science specialist under the surveillance of the Norwegian Animal Research Authority (NhARA) and registered by the Authority (FOTS application ID8017).

**Data Availability Statement:** The data supporting the findings of this study are available from the corresponding author upon reasonable request.

**Acknowledgments:** The authors wish to acknowledge the staff at Sævareid Fiskeanlegg AS for their contribution to this research.

**Conflicts of Interest:** The authors declare no conflict of interest.

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
