# Peer review of "Effect of Photoperiod and Transfer Time on Atlantic Salmon Smolt Quality and Growth in Freshwater and Seawater Aquaculture Systems"

_fishes, doi:10.3390/fishes8040212_

Round 1

Reviewer 1 Report

Dear Authors,

After exhaustive revision I have some comments and suggetions:

Title: need to be changed, where appear "...In Aquaculture protocol"or "aquaculture system" as example

The introduction need to be improve, due to I miss more background about of smoltifitacion in farm, I was checking the scopus I find this paper

"Intestinal incomplete process on the osmoregulation system during Salmo salar smoltification in a productive conditions" Aquaculure 2018, doi.org/10.1016/j.aquaculture.2018.03.022

Other papers where the McCormick describe the photoperiod and T3 effect, thermal effect or combinated effect.

Method: Please to do the graph with sigma plot or another software, The graph will have more quality (as image)

Results

.-need to improve the graphs, for example removed black lines behind the results-lines.

.- To start the graph from "0" and cut the line

.-More information in figure caption, For example n=?, Anova? SEM or SD? etc

I miss other measurements as NAK in kidney or foregut/hindgut, because the gill NKA can be increasing but the other organs are not in the same line.

Discussion

Please to include more information about the Why the weight at final experiment is the same, how was the behaviour in Sea water, the fish keep the scales?

Include more information as introduction comments.

Author Response

Reviewer 1

After exhaustive revision I have some comments and suggestions:

Title: need to be changed, where appear "...In Aquaculture protocol" or "aquaculture system" as example

  • Changed as suggested.

The introduction need to be improve, due to I miss more background about of smoltification in farm, I was checking the Scopus I find this paper

"Intestinal incomplete process on the osmoregulation system during Salmo salar smoltification in a productive conditions" Aquaculture 2018, doi.org/10.1016/j.aquaculture.2018.03.022

Other papers where the McCormick describe the photoperiod and T3 effect, thermal effect or combined effect.

  • We have added more information about the smoltification process in the Introduction.

Method: Please to do the graph with sigma plot or another software, The graph will have more quality (as image).

  • Our graphs are made in the R software that makes scientific graphs with much higher resolution than is possible in Sigma Plot (we have used this program earlier). The quality issue will be easily solved by increasing the pixel size of the each plot.

Results

.-need to improve the graphs, for example removed black lines behind the results-lines.

  • We presume the reviewer is referring to the light grey gridlines used for clarity in Figs. 2-7. We respectfully don’t agree with the reviewer on this point as there is a lot of data on display in those figures and the light grey lines in the background increases the readability and makes tracking data lines easier, not more difficult as it would be if we removed the lines all together.
  • The compromise is to make the lines even lighter and this is something we can do.

.- To start the graph from "0" and cut the line

  • This comment applies to Figures 6 and 7.
  • This is a programming issue in R that shows the bars in the Fig. 6 "hanging", but all start at 0 as can be easily seen in the figure.
  • For Figure 7 all data are shown as mean ± SE (as in all figures) and to increase the readability we allow the figure to start a little bit below the origo. If we had started at origo some of SE bars would not be visible.

.-More information in figure caption, For example n=?, Anova? SEM or SD? Etc

  • This is good suggestion and we have included this information for the figures in question (Figs. 3-7).

I miss other measurements as NAK in kidney or foregut/hindgut, because the gill NKA can be increasing but the other organs are not in the same line.

  • This is a valid comment from the reviewer, but unfortunately, we did not measure the NAK in those organs so that data cannot be included. However, previous investigation in our research group have only found minor differences in NKA between those organs.

Discussion

Please to include more information about the Why the weight at final experiment is the same,

  • This was due to the compensating effect seen in the transfer and photoperiod groups. We have included this topic in the Discussion.

how was the behaviour in Sea water,

  • The behaviour was as expected for this size of salmon after transfer to sea water. We have mentioned this in the Discussion.

the fish keep the scales?

  • We are not sure what the reviewer is referring to here, but all fish showed natural behaviour and there were no damages in their scale at any point during the trial.

Include more information as introduction comments.

  • We have tried to include a short introduction comment at the start of each chapter in the Discussion.

Reviewer 2 Report

The work presented by Martinez, et al. on “Effect of photoperiod and transfer time on Atlantic salmon smolt quality and growth in freshwater and seawater” is interesting and enhance the present understanding of smoltification in relation to photoperiod and subsequent transfer time to seawater on growth and survival. The manuscript is extensive and well written  Few points are mentioned for improvisation of the paper.

My comments for improvement of the MS are noted below:

Abstract:

In line 17, it is mentioned that smoltification is modulated by photoperiod treatment. The fact has not been pronounced in the result. Modify the sentence. The fact that, at the end of the final harvest, there was no significant difference in body weight among the different experimental groups, has to be mentioned in the abstract section. In the abstract, the further research areas in the line may also be included.

Introduction

Line 101: Estimation of appetite is not been mentioned in the paper

Write the extended form of LL, LDN, NKA, etc.

Materials and Methods

Statistical analyses should be crisped and not in the extended manner

Results and Discussion

After How many days of transferring to the seawater, the morality has been started and how long it continued?

Percentage body weight gain may be estimated and mentioned

Line 371-379: The content is repeating and may be made crisped.

As the Final Body weight didn’t effected by photoperiod as well as transfer time, more emphasis may be given on survival aspects. Discuss the effects of the treatments on survival elaborately.

Most of the cited references are old, recent work in this line may be referred in the Introduction and Discussion section. 

Author Response

Reviewer 2

The work presented by Martinez, et al. on “Effect of photoperiod and transfer time on Atlantic salmon smolt quality and growth in freshwater and seawater” is interesting and enhance the present understanding of smoltification in relation to photoperiod and subsequent transfer time to seawater on growth and survival. The manuscript is extensive and well written.

 Few points are mentioned for improvisation of the paper.

Abstract:

In line 17, it is mentioned that smoltification is modulated by photoperiod treatment. The fact has not been pronounced in the result. Modify the sentence. The fact that, at the end of the final harvest, there was no significant difference in body weight among the different experimental groups, has to be mentioned in the abstract section. In the abstract, the further research areas in the line may also be included.

  • We have modified the Abstract according to the comments of the reviewer.

Introduction

Line 101: Estimation of appetite is not been mentioned in the paper

  • This is not correct. The way we estimated the appetite is explained in section 2.3 of the ms. In short food in stomach was assessed 2, 4 and 6 weeks after transfer to seawater (February, March, April and May). This was performed replacing the ordinary diet with an identical diet containing X-ray dense Ballotini glass beads prepared by Skretting (Stavanger, Norway). The new feed had an identical nutrient composition, color, particle size and texture as the normal commercial diet, but it allowed the measurement of individual rates of food intake and stomach evacuation by X-radiography.

Write the extended form of LL, LDN, NKA, etc.

  • We have written the extended forms of all abbreviation first time they are mentioned in the ms.

Materials and Methods

Statistical analyses should be crisped and not in the extended manner

  • We have rewritten this section in a more crisped manner.

Results and Discussion

After How many days of transferring to the seawater, the morality has been started and how long it continued?

  • Mortality was almost absent in all groups apart from the February transfer group. Here the mortality started one week after transfer to seawater and continued for three weeks.

Percentage body weight gain may be estimated and mentioned.

  • We agree with the reviewer and percentage body weight gain SW is shown in Figure 6.

Line 371-379: The content is repeating and may be made crisped.

  • We have rewritten, and shortened, this part.

As the Final Body weight didn’t effected by photoperiod as well as transfer time, more emphasis may be given on survival aspects. Discuss the effects of the treatments on survival elaborately.

  • We agree with the reviewer on this point and have put emphasis on the discussion of survival in the different experimental groups (see section 4.2 on pages 18 and 19).

Most of the cited references are old, recent work in this line may be referred in the Introduction and Discussion section.

  • Yes, some of them are old, but are regarded as "classical" papers in this field of work. We have tried to balance the content using both "classical" (highly cited) papers and more recent works. This is especially evident in the Discussion, whereas the Introduction relays more on the classical literature as it should once introducing the subject.

Reviewer 3 Report

The authors explored the effects of photoperiod regimes and transfer times on Atlantic salmon smolt quality and growth in this study. This study asses how the combination of two photoperiod regimes in freshwater and four transfer times to seawater affected smolt development and subsequent growth in seawater until slaughter during commercial production. The results of this study revealed smoltification and growth in freshwater were only slightly modulated by the photoperiod treatment and much more determined by the limiting effect of the low water temperature during that period. In seawater, growth rate was higher in groups transferred later, which had experienced a delay in growth in freshwater. The study provides some reference information to predict optimal smolt transfer time in industry. The study is addressed and designed moderately. But the authors should provide the information of animal experiment approval issued by Institutional Animal Care and Use Committee in material and methods. Generally, I suggest it can be accepted directly.  

Author Response

Reviewer 3

The authors explored the effects of photoperiod regimes and transfer times on Atlantic salmon smolt quality and growth in this study. This study assess how the combination of two photoperiod regimes in freshwater and four transfer times to seawater affected smolt development and subsequent growth in seawater until slaughter during commercial production. The results of this study revealed smoltification and growth in freshwater were only slightly modulated by the photoperiod treatment and much more determined by the limiting effect of the low water temperature during that period. In seawater, growth rate was higher in groups transferred later, which had experienced a delay in growth in freshwater. The study provides some reference information to predict optimal smolt transfer time in industry. The study is addressed and designed moderately.

But the authors should provide the information of animal experiment approval issued by Institutional Animal Care and Use Committee in material and methods.

  • This is done in Section 2.5 in the M&M. The study was approved by the local representative of Animal Welfare at the Department of Biological Sciences, University of Bergen, Norway (FOTS application ID8017), and samplings were performed as established by the Norwegian Animal Research Authority.

Generally, I suggest it can be accepted directly. 

Round 2

Reviewer 1 Report

I'm agree with authors revision and this Ms can be accepted and published in FISHES.